# Improvement of Surgical View and Working Space at the Lower Pole by Three-Dimensional Exoscope-Assisted Coblation Tonsillectomy: A Case Series

**DOI:** 10.3390/medicina59020259

**Published:** 2023-01-29

**Authors:** Daichi Murakami, Masayoshi Hijiya, Takuro Iyo, Gen Sugita, Sachiko Hayata, Masamitsu Kono, Muneki Hotomi

**Affiliations:** 1Department of Otorhinolaryngology Head and Neck Surgery, Kinan Hospital, 46-70 Shinjo-cho, Tanabe-shi 646-8588, Wakayama, Japan; 2Department of Otorhinolaryngology Head and Neck Surgery, Wakayama Medical University, 811-1 Kimiidera, Wakayama-shi 641-5810, Wakayama, Japan; 3Gen ENT Clinic, 3-14-1, Takasu, Mihama-ku, Chiba-shi 261-0004, Chiba, Japan

**Keywords:** exoscope, coblation, tonsillectomy

## Abstract

Advantages of hot devices for tonsillectomy, represented by coblation, have been highlighted in recent years. During hot technique tonsillectomy it is important to identify and coagulate the vessels of the tonsillar capsule, especially at the lower pole of the tonsil. Hot technique tonsillectomy under microscope or endoscope has therefore been recommended to achieve accurate identification of the microstructure of the surgical field. We have applied ORBEYE, a three-dimensional surgical exoscope system, to coblation tonsillectomy. Advantages of using ORBEYE include high definition and high magnification images, and flexibility of camera position and angle. This means there is an improved surgical view and working space, particularly at the lower pole during performance of coblation tonsillectomy. Here, we demonstrate that ORBEYE can be an effective surgical instrument in coblation tonsillectomy.

## 1. Introduction

Tonsillectomy is one of the most common surgeries in otorhinolaryngology, head and neck surgery. In addition to surgical procedures performed by steel knife or guillotine, the so-called ‘cold method’, some new techniques, known as ‘hot technique’ have been developed and include electrocautery, harmonic scalpel, bipolar diathermy dissection, radiofrequency and coblation. Hot techniques have improved the outcome of tonsillectomy, including achieving lower intraoperative blood loss, shorter operation time and lower postoperative pain [1]. However, degenerative color changes after tissue coagulation in hot techniques sometimes create difficulty in making detailed anatomical identification of the areas surrounding the tonsillar capsule [2]. Narrow surgical view during tonsillectomy makes it more difficult to dissect tonsils in the appropriate layer, especially at the lower pole of the tonsils [3]. Adequate coagulation of vessels at the lower pole is reported to be an important factor in avoiding postoperative hemorrhage [2,4]. However, complete coagulation is sometimes difficult because of the abundance of small vessels entering the lower pole that require frequent coagulation, and obstruction of the surgical view by the root of the tongue being closely attached to the lower pole [2,4,5].

The utility of surgical microscopes and endoscopes for precisely identifying anatomy in hot technique tonsillectomy has been reported [3,6,7,8,9]. The three-dimensional (3D) surgical exoscope has, for example, been applied in the field of otorhinolaryngology head and neck surgery [10,11,12]. Notably, in transoral pharyngolaryngeal surgery, use of the surgical exoscope has advantages compared with conventional microscopic or endoscopic surgery, including high definition and high magnification 3D images, flexibility of camera position and angle, and wide working space [11].

We have applied coblation tonsillectomy under a surgical microscope, and introduced ORBEYE, a 3D surgical exoscope system, to coblation tonsillectomy. Here, we report the advantages of ORBEYE in improving the surgical view and working space, particularly in the lower pole of the tonsils, during coblation tonsillectomy in both pediatric and adult patients. To our knowledge, this is the first clinical report on the utility of ORBEYE-assisted coblation tonsillectomy.

## 2. Case Presentation

This is the first case series of three sequential patients that underwent coblation tonsillectomy under ORBEYE performed by a single surgeon (D.M.). Details of the patients are summarized in Table 1. Surgical indication for two patients was obstructive sleep apnea (OSA), and one patient for recurrent tonsillitis. Bilateral tonsillectomy was performed in patient No. 1 and 3, and bilateral tonsillectomy with adenoidectomy in patient No. 2. Surgical view at the lower pole was scored according to the Wang classification system [4]. All procedures were performed under general anesthesia. All patients and their guardians agreed to publish this paper. Written informed consent was obtained from the two adult patients and the parent of the pediatric patient. The study was conducted according to the guidelines of the Declaration of Helsinki, and approved by the Institutional Review Board of Kinan hospital (protocol code; 253).

### 2.1. Equipment and Operating Room Setup

Figure 1 shows the operating room setup.

The ORBEYE 3D surgical exoscope system (Olympus Corporation, Tokyo, Japan), with 4K-3D display, Coblator II Surgery System (Smith & Nephew KK, London, UK) and Evac70 70 EXTRA HP (Smith & Nephew KK) were used. The operator took position in front of the head of the patient and wore 3D glasses. The main unit of ORBEYE was placed to the patient’s left side, and the camera arm was introduced from the operator’s left side. As the main unit of ORBEYE is smaller than or as small as those of a microscope, it was possible to place ORBEYE without restricting sets of other devices in the operating room. The wireless footswitch of ORBEYE was placed at the left foot of the operator. To allow the operator to perform transoral procedures while looking straight ahead, the 4K-3D monitor was placed at the patient’s feet. The preparation was as simple as covering the exoscope with a drape, and operation could be started immediately. The coblator was placed to the patient’s right, and the foot pedal was placed for use by the operator’s right foot. The scrub nurse stood on the left side because, for the right-handed operator, only the coblator wand was used with the right hand, and forceps or other instruments were used with the left hand. An anesthesiologist stood to the operator’s right side.

### 2.2. Surgical Procedure

The anterior palatine arch was incised by ablation mode 7. Connective tissue adhering to the tonsil was ablated along the capsule (Figure 2a). The vessels entering the tonsils were dissected after prior coagulation in coagulation mode 3. The high definition and high magnification 3D images and flexible camera position provided by ORBEYE enabled early detection and coagulation of the vessels and an excellent surgical view (Figure 2b,c), regardless of the operator’s position. ORBEYE especially provides excellent clear and wide surgical views of the lower pole of tonsil, where the root of the tongue is closely attached and disturbs the surgical views. In all three cases, a grade 1 view of the lower pole [4] was achieved during the procedure. Only the first case did it take more time to get used to manipulating ORBEYE. However, after the first case, we achieved a better learning curve by carrying out enough experiments to perform ORBEYE-assisted tonsillectomy. The learning curve of ORBEYE-assisted tonsillectomy depends on the surgical procedures of tonsillectomy such as cold and hot methods rather than the manipulation of the system.

Furthermore, a wide working space was obtained, even during dissection of the lower pole (Figure 3a,b). Typically, the cramped position of the operator, with the head low and looking up, or interference between the coblator wand and the tongue root or microscope make for unsatisfactory working space [2]. After hemostasis in the tonsillar bed, the surgical procedure was completed. Throughout the procedure, we observed very little bleeding and the intraoperative blood loss is estimated at less than 1 mL per patient (Table 1). In the three cases, it took about one hour (60 to 74 min) to perform ORBEYE-assisted tonsillectomy. The durations were similar to those of tonsillectomy in past cases under the surgical microscope. There were no postoperative complications including postoperative hemorrhage in any of the patients (Table 1).

## 3. Discussion

In tonsillectomy, difficulty and complications can depend on surgical view and working space, especially at the lower pole [2,4,5]. Inadequate recognition of the boundary between the tonsillar lower pole and the tongue root reportedly increases the risk of impairment of the lingual artery and a branch of the tonsillar artery [2,4]. This is the first report on the advantages of 3D-exoscope ORBEYE-assisted coblation tonsillectomy with attention to surgical view and working space. We reported the performance of precise surgical technique in 3D-exoscope-assisted coblation tonsillectomy.

ORBEYE facilitated excellent surgical view of the lower pole of the tonsils. Risk factors of postoperative hemorrhage include age of the patient and the level of experience of the surgeon [5]. Wang et al. suggested the anatomical diversity of the tonsils as a further risk factor of postoperative hemorrhage [4]. Poor surgical view of the tonsillar lower pole attached to the tongue root is related to higher occurrence of postoperative hemorrhage [4]. To reduce postoperative hemorrhage, it is important to improve the surgical view of the tonsil, especially at the tonsillar lower pole. Previous reports have indicated the utility of surgical microscopes [3,6] and endoscopes [7,8,9]. However, surgical microscopes provide 3D surgical view for the surgeon only. Furthermore, the field and angle of view are limited because the operator’s view of the surgical field is via an eyepiece [7]. Meanwhile, the surgical view in endoscopic tonsillectomy is much closer and it is easier to adjust the angle, but the visual image is not 3D [3]. Further, surgeons must take care to avoid unexpected injury to the face and lips because they are not included in the enlarged endoscopic view [7]. In our cases, ORBEYE provided detailed surgical view, especially at the lower pole. The higher definition and higher magnification of 3D image and the flexibility of position of the camera enabled an excellent view of surgical fields, irrespective of the operator’s position. Grade 1 views of the lower pole, as defined by Wang et al. [4], were achieved in all three cases. Fine adjustment of the surgical view and a change of magnification are possible with a foot switch, meaning surgery is not interrupted.

Excessive hemorrhage in the surgical field is one of the anxious dificulties during tonsillectomy. It is hard to compare with other devices, as exessive hemorrhage is not a frequent complication among simple adenoid vegetation and tonsillar hypertrophy. However, based on the following points, ORBEYE would be useful even in cases of exessive bleeding. In cases of exessive hemorrhage, if a microscope is used, the operator needs to remove the blood with low magnification, detect the bleeding points, and then perform a rapid hemostasis with high magnification. In contrast, an exoscope can provide high definition and high magnification 3D images even in cases of excessive hemorrhage. In addition, ORBEYE can provide instant switching of the angle of view and the magnification by a footswitch, resulting in an efficient hemostasis. Another type of exoscope represented by VITOM (Karl Storz, Tuttlingen, Germany) is fixed by the pneumatic holder, so the operator needs to manually mobilize the microscope and adjust zoom and focus [13]. These result in the interruption of surgical procedure. Unlike an endoscope, the exoscope allows the surgeon to view the surgical field from outside the oral cavity. The possibility of blood adhering to the scope and interfering with the field of view is greatly reduced. In addition to the clear magnified view to identify the bleeding point under exoscope, coblator wand having both suction port and hemostatic bipolar makes it easy to remove hemorrhages as well as hemostasis.

A wider working space could be obtained under ORBEYE. In microscopic tonsillectomy, the operator must stay in a position with their head low and looking up during surgical procedures at the lower pole of the tonsils. Because ORBEYE has an extended focal length of 220 to 550 mm, the lens of the exoscope can be positioned away from the face of the patient. It enables avoidance of interference between the coblator wand and the exoscope. Interference between the long shaft of the coblator wand and the microscope can limit the working space. Interference between the coblator wand and endoscope is inevitable in endoscopic tonsillectomy, in which the endoscope must be positioned inside the patient’s mouth [14]. In addition, the endoscopic method requires not only a surgeon, but also a well-trained camera operator to prevent the interference between the device and the endoscope. Magnification capacity and angle flexibility facilitated by ORBEYE prevented such interference between the surgical devices and allowed the operator a wider working space.

We became familiar with ORBEYE-assisted tonsillectomy easily after the introduction. In the field of neuroendoscopic surgery, there are some reports regarding the impact of exoscope on the learning curve. Parihar et al. reported that the learning curve is found initially in pure endoscopic procedures and the video telescope operating monitor system, which is a type of exoscope that effectively reduces the initial learning curve of neuroendoscopy [15]. In these present cases, the first case for which the ORBEYE was introduced required more time to become accustomed to manipulation of the exoscope. However, from the second case we became easily accustomed to the manipulation of ORBEYE. The duration of surgery in each case was the same as the duration of our previous experience of tonsillectomy under the microscope. As long as the surgeon is well trained for adenotonsillectomy, ORBEYE becomes easy to adapt to after the introduction, because ORBEYE has characteristics similar to both microscope (three-dimensional image) and endoscope (easy to adjust angle of view). Based on these basic characteristics of ORBEYE, we suspect the learning curve would be mandatory to get the right impression of this tool faster. We judged the ORBEYE to be an easy-to-use exoscope.

ORBEYE would have some advantages over other exoscopes. Langer et al. recently reported a review about the comparison among some types of exoscope [13]. In the report, four exoscopic systems used in neurosurgery—VITOM, KINEVO (Carl Zeiss AG, Oberkochen, Germany), Modus V, and ORBEYE (Olympus, Tokyo, Japan)—were reviewed according to the individual authors’ experiences. Although VITOM has similar characteristics to ORBEYE, the operator needs to manipulate pneumatic camera arm manually as mentioned above. The advantage of KINEVO is the point-lock function (locking the focal point to allow movement of the head around that fixed point for alternative views), however, its heavy footprint/size occupies large space. Modus V provides only 2D visual image. ORBEYE can provide 3D visual images and be operated with a footswitch without interruption of the two-handed procedure. ORBEYE would be a suitable exoscope for coblation tonsillectomy because coblation tonsillectomy requires a continuous two-handed procedure and space to set various devices including main unit of the coblator.

The current report focused on a limited case series without a control group. As the current report is a case series that focused on the introduction of ORBYEYE for adenotonsillectomy using a powered device, it has no control group. To show further safety and feasibility, accumulation of cases and future studies about comparisons between exoscopes and other conventional devices such as microscopes and endoscopes are needed. It is also better to evaluate which surgical techniques (e.g., cold method, coblation and other powered devices) show better compatibility with ORBEYE in future studies. However, we can deduce that this first case series has value with respect to discussing and indicating the utility of ORBEYE-assisted coblation tonsillectomy. The costs for installing ORBEYE (all components including exoscope unit and monitor) is about the same as those for microscopes. Running costs, such as the cost of surgical drapes, are comparable to those of microscopes. Although the initial costs are expensive, it is considered that ORBEYE provides clear surgical view and wide working space contributing to safe and stressless surgery, and can be applicable for various surgeries in otolaryngologic and head and neck surgery field including otologic surgery, laryngomicrosurgery, and transoral pharyngolaryngeal surgery. The introduction of ORBEYE will have great significance. In addition to the advantages of ORBEYE regarding its superiority over microscopes and endoscopes, its use has significant educational value in surgical training. Instructors and senior surgeons can show residents surgical procedures without interruption and changing positions, and conversely, they can monitor residents’ surgical performance with an extremely fine view.

## 4. Conclusions

ORBEYE, a high-resolution 3D surgical exoscope system, had various advantages including high definition and high magnification 3D visual images and flexibility of camera position and angle, which provided excellent surgical view and working space in coblation tonsillectomy from early on after the introduction. ORBEYE especially worked well during the procedure at the lower pole of tonsil. Although accumulation of cases and future studies about comparison between exoscope and other conventional devises such as microscope and endoscope are needed, through our three cases, we demonstrated the advantages of ORBEYE-assisted coblation tonsillectomy. The ORBEYE-assisted tonsillectomy is an attractive alternative surgical procedure to conventional procedures. The advantages of ORBEYE should be further evaluated in randomized cohort studies.

## Figures and Tables

**Figure 1 medicina-59-00259-f001:**
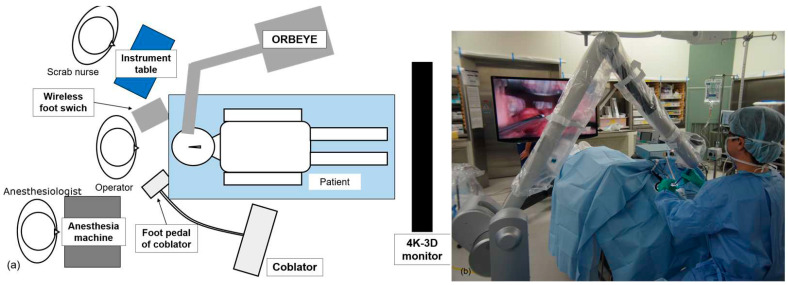
Equipment and operating room setup during exoscope-assisted coblation tonsillectomy. (**a**) Illustration of operating room setup. (**b**) Actual setup in the operation room.

**Figure 2 medicina-59-00259-f002:**
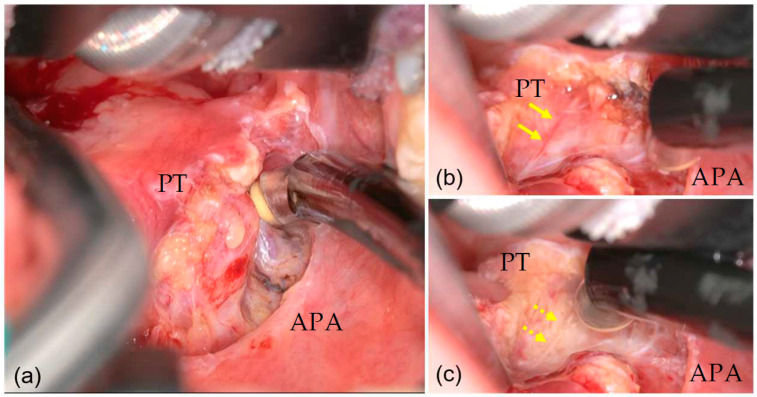
Surgical view under ORBEYE (right side). (**a**) High definition and high magnification surgical views were available. (**b**) Small vessel entering the lower pole (arrow) were detected. (**c**) Coagulation of the vessel (dashed arrow) was completed. PT: indicates palatine tonsil; APA: anterior palatine arch.

**Figure 3 medicina-59-00259-f003:**
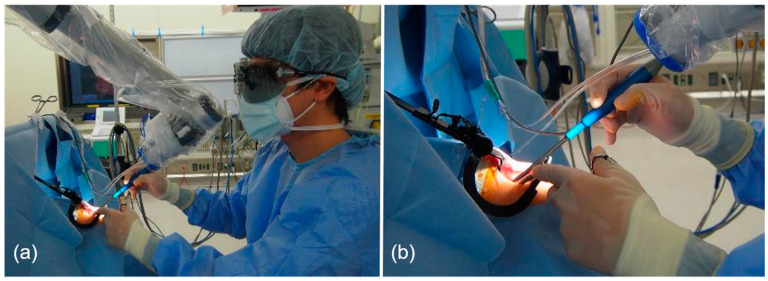
Positions of the camera and the operator. (**a**) The operator had a wide working space and relaxed head-up position. (**b**) There was no interference between the exoscope and the coblator wand.

**Table 1 medicina-59-00259-t001:** Patient characteristics.

Patient No.	1	2	3
Sex	Female	Male	Male
Age (years)	56	6	38
Surgical indication	OSA	OSA	RT
Type of surgery	T	T & A	T
View of lower pole [4]	Grade 1	Grade 1	Grade 1
Operation time (min)	65	60	74
Blood loss (mL)	<1	<1	<1
Complications	None	None	None

OSA: indicates obstructive sleep apnea; RT: recurrent tonsillitis; T: tonsillectomy; T & A: tonsillectomy with adenoidectomy.

## Data Availability

The data presented in the present study are available on request from the corresponding author.

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
