# Peer review of "Improvement of Surgical View and Working Space at the Lower Pole by Three-Dimensional Exoscope-Assisted Coblation Tonsillectomy: A Case Series"

_medicina, 2023, doi:10.3390/medicina59020259_

Round 1

Reviewer 1 Report

The authors write about their experiences with the ORBEYE for dissection of the lower pole of the tonsils. It is is as an intersting topic for ENT surgeons. The authors state, that they demonstrated the advantages of ORBEYE assisted tonsillectomy. But there is no comparison with another tool such as the microscope or endoscope or regular technique. It is more a description that "it works" rather than that it works better. There is no control group and they only write about three cases. 

Furthermore information about operation time, costs, preparation, learning curve what be mandatory to get the right impression of this tool. 

Therefore the manuscript needs revisions before it can be considered for publication. 

Author Response

Response to Reviewer 1 Comments

We appreciate the editor and reviewer’s careful review and thoughtful feedback. We agree with your suggestion for modification of our manuscript. Here we made the answer to the comments below and modified the manuscript and table.

We attach the revised version of manuscript. With these changes to our final manuscript, we hereby resubmit the manuscript for a secondary evaluation. Thank you once again for your consideration of our paper.

Point 1: The authors state, that they demonstrated the advantages of ORBEYE assisted tonsillectomy. But there is no comparison with another tool such as the microscope or endoscope or regular technique. It is more a description that "it works" rather than that it works better. There is no control group and they only write about three cases.

Response 1: Thank you for the valuable comment. The reviewer is correct. As the reviewer points out, this is ‘case series’ of only three cases not the comparative study between exoscope and other methods. Although future studies about comparison between exoscope and other conventional devises such as microscope, endoscope or direct vision are needed, we can introduce that this first case series will have enough value to discuss and indicate the utility of ORBEYE assisted coblation tonsillectomy. We mentioned these points as limitation paragraph in Discussion part (line 169-173) according to the reviewer’s comment.

Point 2: Furthermore, information about operation time, costs, preparation, learning curve what be mandatory to get the right impression of this tool.

Response 2: Thank you for your comment. The reviewer is correct.

We added operation time to the table (line 71). Because operation time varied according to types of surgery (tonsillectomy or tonsillectomy with adenoidectomy) we also added types of surgery to Case Presentation part (line 59-61 and line 73-74) and table (line 69).

The costs for installing ORBEYE (all components including exoscope unit and monitor) is about the same as those for microscope. Running costs, such as cost of surgical drape, are comparable to those of a microscope. Although the initial costs are expensive, it is considered that ORBEYE is a preferable instrument to microscopes and endoscopes, and can be used in other otolaryngologic and head and neck surgical procedures including otologic surgery and transoral pharyngolaryngeal surgery. The introduction of ORBEYE will have great significance. We mentioned these points as limitation paragraph in Discussion part (line 173-180).

The preparation was simple as covering exoscope by drape, and operation could be immediately started. We mentioned these points in 2.1. Equipment and Operating Room Setup paragraph in Case Presentation part (line 90-91).

We could be familiar to ORBEYE assisted tonsillectomy early after the introduction. ORBEYE has characteristics similar to both microscope (three-dimensional image) and endoscope (easy to adjust angle). Based on these basic characteristics of ORBEYE, we suspect the learning curve of ORBEYE will be faster. We mentioned these points in Discussion part (line 165-168).

Reviewer 2 Report

First of all, I want to congratulate the authors for their work.

Here are my comments:

1) This study present the advantages of using an exoscope in narrow space(tonsillectomy). Several cases will show the safety and feasibility of using this device.

2) The authors should mention the type of anesthesia used for this procedures.

I will expect some comments regarding the possible difficulties of access in surgical field in case of massive hemorrhage.

Author Response

Response to Reviewer 2 Comments

We appreciate the editor and reviewer’s careful review and thoughtful feedback. We agree with your suggestion for modification of our manuscript. Here we made the answer to the comments below and modified the manuscript and table sent to the editor.

We attach the revised version of manuscript. With these changes to our final manuscript, we hereby resubmit the manuscript for a secondary evaluation. Thank you once again for your consideration of our paper.

Point 1: This study present the advantages of using an exoscope in narrow space (tonsillectomy). Several cases will show the safety and feasibility of using this device.

Response 1: Thank you for the valuable comment. The reviewer is correct. As you pointed out, the current report focused on the limited case series without control group. To show further safety and feasibility, accumulation of cases and future studies about comparison between exoscope and other conventional devises such as microscope and endoscope are needed. However, we can introduce that this first case series will have enough value to discuss and indicate the utility of ORBEYE assisted coblation tonsillectomy. We mentioned these points as limitation paragraph in Discussion part (line 169-173) according to the reviewer’s comment.

Point 2: The authors should mention the type of anesthesia used for this procedures.

Response 2: Thank you for your pointing for the type of anesthesia. We apologize for not describing the type of anesthesia and added that ‘All procedures were performed under general anesthesia‘ to Case Presentation part (line 62).

Point 3: I will expect some comments regarding the possible difficulties of access in surgical field in case of massive hemorrhage.

Response 3: Thank you for the valuable comment. The reviewer is correct that massive bleeding in the operative field is a distressing problem in tonsillectomy. However, exoscope can provide the high definition and high magnification 3D images even in cases with massive hemorrhage. Unlike endoscope, the exoscope allows the surgeon to view the surgical field from outside the oral cavity. The possibility of blood adhering to the scope and interfering with the field of view is greatly reduced. In addition to the clear magnified view to identify the bleeding point under exoscope, coblator wand having suction can easy to remove hemorrhage as well as hemostasis. We added the statement about the situation of massive hemorrhage in Discussion part (line 148-154).

Round 2

Reviewer 1 Report

Thank you for your adjustements of the paper according to the reviewers comments. I can accept the paper in the current form.